# Colesevelam Reduces Ethanol-Induced Liver Steatosis in Humanized Gnotobiotic Mice

**DOI:** 10.3390/cells10061496

**Published:** 2021-06-14

**Authors:** Noemí Cabré, Yi Duan, Cristina Llorente, Mary Conrad, Patrick Stern, Dennis Yamashita, Bernd Schnabl

**Affiliations:** 1Department of Medicine, University of California San Diego, La Jolla, CA 92093, USA; ncabrecasares@ucsd.edu (N.C.); yid003@ucsd.edu (Y.D.); allorenteizquierdo@health.ucsd.edu (C.L.); 2Axial Therapeutics, Woburn, MA 01801, USA; mary@axialtx.com (M.C.); patrick@axialtx.com (P.S.); yamashitad@yahoo.com (D.Y.); 3Department of Medicine, VA San Diego Healthcare System, San Diego, CA 92161, USA

**Keywords:** alcoholic liver disease, bile acids, CYP7A1, microbiome

## Abstract

Alcohol-related liver disease is associated with intestinal dysbiosis. Functional changes in the microbiota affect bile acid metabolism and result in elevated serum bile acids in patients with alcohol-related liver disease. The aim of this study was to identify the potential role of the bile acid sequestrant colesevelam in a humanized mouse model of ethanol-induced liver disease. We colonized germ-free (GF) C57BL/6 mice with feces from patients with alcoholic hepatitis and subjected humanized mice to the chronic–binge ethanol feeding model. Ethanol-fed gnotobiotic mice treated with colesevelam showed reduced hepatic levels of triglycerides and cholesterol, but liver injury and inflammation were not decreased as compared with non-treated mice. Colesevelam reduced hepatic cytochrome P450, family 7, subfamily a, polypeptide 1 (Cyp7a1) protein expression, although serum bile acids were not lowered. In conclusion, our findings indicate that colesevelam treatment mitigates ethanol-induced liver steatosis in mice.

## 1. Introduction

Alcohol abuse is one of the most important causes for liver disease worldwide [1]. Alcohol-related liver disease includes steatosis, fibrosis, cirrhosis, and alcoholic hepatitis [2]. Alcoholic hepatitis is the most severe form of alcohol-related liver disease with mortality ranging from 20% to 50% at 28 days and up to 70% at 90 days [3,4]. Current therapies for alcoholic hepatitis remain limited to corticosteroids, however, almost half of patients cannot tolerate or do not respond to corticosteroid therapy [5,6]. The only intervention shown to improve long-term survival in alcohol hepatitis is alcohol abstinence, but recidivism is high [7,8]. Early liver transplantation is the only curative therapy [9,10], but there is limited acceptance of early transplantation as the standard of care treatment.

The gut–liver axis is important for the progression of alcohol-associated liver disease in both patients and experimental models [11,12]. Previous studies have confirmed the link between alterations in the microbiota composition and progression of alcohol-related liver disease [13,14]. In particular, the colonization of germ-free mice with stool from patients with alcoholic hepatitis increases ethanol-induced liver disease as compared with conventional mice [15].

Bile acids produced from cholesterol in the liver and metabolized by enzymes encoded by the gut microbiota are critical for maintaining a healthy gut microbiota [11]. Bile acid synthesis is tightly regulated by negative feedback inhibition through the nuclear receptor farnesoid X receptor (FXR)–fibroblast growth factor 15/19 (FGF15/19) axis that also modulates the gut barrier and is an integral part of the gut–liver axis [13]. Changes in the microbiota composition promote the deconjugation of primary bile acids and the accumulation of bile acids in the liver and in the systemic circulation. Increased bile acids in the liver can result in hepatocellular damage followed by inflammation and fibrosis [16].

Bile acid sequestrants, such as colesevelam, are ion exchange polymers that bind bile acids in the gut. Colesevelam inhibits bile acid reabsorption, leading to increased bile acid synthesis and reduced cholesterol levels in patients with cholestatic pruritus and Crohn’s disease [17,18]. Colesevelam is indicated as an adjunct therapy to diet and exercise to reduce elevated low-density lipoprotein cholesterol (LDL-C) in patients with hypercholesterolemia and glycemic control in adults with type 2 diabetes mellitus [19,20,21]. Interestingly, colesevelam reduced serum bile acid levels, serum liver enzymes, and the expression of inflammatory and fibrotic markers in a mouse model of sclerosing cholangitis (multidrug resistance protein 2 (*Mdr2*)^−/−^ mice) [22], indicating potential anti-cholestatic and bile duct-protective properties.

The aim of this study was to identify the potential therapeutic role of the bile acid sequestrant colesevelam in a humanized mouse model of ethanol-induced liver disease (NIAAA model) [23,24].

## 2. Materials and Methods

### 2.1. Mice

Female C57BL/6 germ-free mice were bred at UC San Diego and used in this study. Fecal transplantation with a stool sample from one individual patient with alcoholic hepatitis was performed at the age of 5–6 weeks and repeated 2 weeks later, as previously described [23]. A total of two patients with alcoholic hepatitis were chosen as stool donors (Table 1).

Briefly, mice were gavaged each time with 100 μL of stool samples (1 g stool dissolved in 30 mL Luria–Bertani (LB) medium containing 15% glycerol under anaerobic conditions). Two weeks after the second gavage, mice were placed on a chronic-binge ethanol diet (NIAAA model), as previously described [24]. Mice were fed with Lieber–DeCarli diet and the caloric intake from ethanol was 0% on days 1–5 and 36% from day 6 until the end of the study period. At day 16, mice were gavaged with a single dose of ethanol (5 g/kg body weight) in the early morning and sacrificed 9 h later. Pair-fed control mice received a diet with an isocaloric substitution of dextrose (Figure 1).

Colesevelam 2% (Apothecon Pharmaceuticals Pvt. Ltd. Vadodara, Gujarat, India), based on dry powder weight, was added to the liquid diet from day 6 until the end of the study period, both in the control and ethanol-containing diet groups. To keep compounds in the liquid diet, xanthan gum (0.4% based on weight) was added to the liquid diet. Initial experiments confirmed that increased concentrations of xanthan gum did not affect the daily amount of consumed ethanol-containing liquid diet. Mice were randomly assigned into groups at the beginning of the study. One mouse in each of the two groups (vehicle and colesevelam) died during the ethanol feeding period.

All animal studies were reviewed and approved by the Institutional Animal Care and Use Committee of the University of California, San Diego (protocol number: S09042).

### 2.2. 16S rRNA Sequencing

The fecal DNA extraction, 16S rRNA sequencing, and analysis of stool samples from two patients with alcoholic hepatitis have been reported before [23]. Raw 16S sequence reads can be found in the NCBI SRA associated with Bioproject PRJNA525701.

### 2.3. Real-Time Quantitative PCR

RNA was extracted from liver tissues using Trizol (Invitrogen, Carlsbad, CA, USA). RNA was digested with DNase using the DNA-free DNA removal kit (Invitrogen, Carlsbad, CA, USA), and cDNAs were generated using the high-capacity cDNA reverse transcription kit (Applied Biosystems, Foster City, CA, USA). Primer sequences for mouse genes were obtained from the NIH qPrimerDepot. All primers used in this study are listed in Table 2. Mouse gene expression was determined with Sybr Green (Bio-Rad Laboratories, Hercules, CA, USA) using ABI StepOnePlus real-time PCR system. The qPCR value of the gene of interest expression was normalized to the total amount of 18S as housekeeping gene.

### 2.4. Staining Procedures

Formalin-fixed tissue samples were embedded in paraffin (Paraplast plus, McCormick, St. Lous, MO, USA) and stained with hematoxylin-eosin (Leica Biosystems Inc., Buffalo Grove, IL, USA). To determine lipid accumulation, liver sections were embedded in OCT compound. Next, 8-μm frozen sections were cut and stained with Oil Red O (Sigma-Aldrich, St. Louis, MO, USA).

### 2.5. Biochemical Analysis

Serum levels of alanine aminotransferase (ALT) were measured at 340 nm using Infinity ALT kit (Thermo Scientific, San Diego, CA, USA). The assay is linear up to 450 U/L. Serum alkaline phosphatase (ALP) was measured using QuantiChrom Alkaline Phosphatase Assay Kit (BioAssay Systems, Hayward, CA, USA) at 405 nm at time 0 and time 4. Levels of ethanol were measured using the Ethanol Assay Kit (BioVision, Milpitas, CA, USA) at 570 nm. Hepatic triglyceride levels were measured using the Triglyceride Liquid Reagents Kit (Pointe Scientific, Canton, MI, USA). The procedure is linear up to 1000 mg/dL and was measured at 500 nm. Hepatic cholesterol levels were measured at 600 nm using the Cholesterol Liquid Reagents Set (Pointe Scientific, Canton, MI, USA). Serum lipopolysaccharides (LPS) was measured by ELISA (Lifeome Biolabs, Oceanside, CA, USA). The minimum detectable dose of LPS is 5.15 ng/mL. Total serum bile acids were measured using a Mouse Total Bile Acid kit (Crystal Chem, Elk Grove Village, IL, USA) at 540 nm.

### 2.6. Immunoblot Analyses

To measure the expression of Cyp7a1 (also called cholesterol 7α-hydroxylase), protein was extracted from frozen hepatic tissues (20 mg) and homogenized with 200 µL RIPA (Thermo Scientific, San Diego, CA, USA) with phosphatase inhibitors (Sigma Aldrich, St. Louis, MO, USA) and protease inhibitors (Sigma Aldrich, St. Louis, MO, USA). The proteins were resolved on 8% polyacrylamide gels by SDS-PAGE and transferred to polyvinylidene difluoride membranes (Bio-Rad Laboratories, Hercules, CA, USA). Immunoblot analysis was performed using anti-Cyp7a1 antibody (1:1000; ab65596, Abcam). Anti-tubulin antibody (1:1000; sc-5286, Santa Cruz Biotechnology, USA) was used to ensure equal loading for liver tissue. Densitometry was done using Image Lab 2.0 software (Bio-Rad Laboratories, Hercules, CA, USA).

### 2.7. Statistical Analysis

Results are expressed as mean ± s.e.m. Numbers for biological replicates are *n* = 5 for control diet and *n* = 14–15 for ethanol-containing diet groups. Two technical replicates were performed in the control diet-fed groups; five technical replicates were performed in the ethanol diet-fed groups. Significance was evaluated using one or two-way analysis of variance (ANOVA) with Tukey’s post-hoc test. A *p* value < 0.05 was considered to be statistically significant. Statistical analyses were performed using R statistical software, R version 1.3.1093, 2020, the R Foundation for Statistical Computing, and GraphPad Prism v8.4.3

## 3. Results

### 3.1. Colesevelam Reduces Ethanol-Induced Liver Steatosis in Gnotobiotic Mice

To determine the effect of colesevelam in a mouse model of ethanol-induced liver disease, germ-free C57BL/6 mice were colonized with stool from two patients with alcoholic hepatitis (Table 1) and subjected to the chronic-binge ethanol feeding model [23].

The 16S rRNA analysis of stool from the two patients with alcoholic hepatitis is shown in Figure 2a. Colesevelam treatment did not affect body or liver weight in control or ethanol-fed groups. Daily food intake was similar in pair-fed mice administered colesevelam and fed a control or ethanol-containing diet (Figure 2b–d). Although colesevelam reduced hepatic expression of two enzymes that metabolize ethanol in the liver of ethanol-fed mice, alcohol dehydrogenase 1 (*Adh1*) and cytochrome P450 family 2 subfamily e polypeptide 1 (*Cyp2e1*), this did not significantly change the serum level of ethanol (Figure 2e–g).

Mice developed liver injury, as indicated by an increased serum level of ALT and ALP (Figure 3a,b), and liver steatosis, as evidenced by increased hepatic levels of triglycerides and cholesterol, following ethanol feeding (Figure 3c–d). Mice treated with colesevelam showed higher serum ALT and ALP levels as compared with vehicle mice following chronic ethanol feeding (Figure 3a,b). Colesevelam reduced ethanol-induced liver steatosis, as shown by lower levels of hepatic triglycerides and cholesterol (Figure 3c–d). No significant differences were detected in liver phenotype, including liver weight, serum ethanol and ALT, and hepatic triglycerides and cholesterol, between ethanol-fed mice colonized with stool from the two donors (not shown). Hepatic genes involved in de *novo* lipogenesis (acetyl-coenzyme A carboxylase *(Acc)-1* and fatty acid synthase (*Fasn*)), and cholesterol synthesis (sterol regulatory element-binding protein (*Srebp*)-*1c* and 3-hydroxy-3-methylglutaryl-coenzyme A reductase (*Hmgcr*)) were lower in ethanol-fed mice treated with colesevelam (Figure 3e–h). Histological analysis confirmed that ethanol feeding increased lipid droplets in H&E- and Oil Red O-stained liver sections, while colesevelam decreased lipid droplet accumulation following ethanol feeding (Figure 3i,k). These results indicate that colesevelam treatment reduces ethanol-induced liver steatosis.

### 3.2. Effect of Colesevelam Treatment on Hepatic Inflammation

Hepatic expression of inflammatory cytokines and chemokines, including chemokine (C-X-C motif) ligand 1 (*Cxcl1*), *Cxcl2*, and chemokine C-C motif ligand 2 (*Ccl2*), was not significantly different in mice treated with colesevelam following chronic ethanol feeding (Figure 4a–c). However, myeloperoxidase (*Mpo*), marker of neutrophil azurophilic granules, was significantly decreased after treatment with colesevelam (Figure 4d).

### 3.3. Colesevelam Treatment and Bile Acids

Intestinal permeability increased in ethanol-fed mice compared with mice fed with an isocaloric diet, as shown by higher levels of serum LPS. Colesevelam did not affect intestinal barrier function in ethanol-fed mice (Figure 5a), indicating that it did not reverse ethanol-induced gut barrier dysfunction.

Colesevelam reduced the hepatic expression of Cyp7a1 protein (Figure 5b-c) and *Cyp27a1* mRNA expression (Figure 5d), while hepatic *Cyp7b1* and *Cyp8b1* mRNA expression were unchanged in mice fed chronic-binge ethanol diet (Figure 5e-f). The serum levels of bile acids were similar in ethanol-fed mice treated with vehicle or colesevelam (Figure 5g).

## 4. Discussion

Alcohol-related liver disease has been among the leading causes of cirrhosis and liver-related death [2]. Here, we demonstrated that colesevelam reduces liver steatosis, but not liver injury and inflammation in ethanol-fed gnotobiotic mice. This was associated with reduced expression of hepatic Cyp7a1, the rate-limiting enzyme in bile acid synthesis.

Patients with alcoholic hepatitis have significantly higher levels of serum bile acids in relation to controls and patients with alcohol use disorder [13]. Treatment strategies to improve bile acid homeostasis may prevent or slow the progression of liver disease [14]. The intestine-restricted FXR agonist fexaramine protected mice from ethanol-induced liver injury, steatosis, and inflammation [14]. While bile acid metabolism was only minimally altered, fexaramine stabilized the gut barrier and improved hepatic lipid metabolism [14]. Similarly, a non-tumorigenic FGF19 variant, a human FGF15 ortholog, showed a strong beneficial metabolic effect in the liver, ameliorated ethanol-induced steatohepatitis, but did not lower systemic bile acid levels [14].

In the present study, instead of directly modulating FXR/FGF19/15 signaling, we aimed at the intestinal sequestration of bile acids. Bile acid sequestrants, such as cholestyramine, colesevelam, and colestipol, are positively charged molecules that bind negatively charged intestinal bile acids to block bile acid reabsorption and cholesterol absorption [25], leading to the increased conversion of cholesterol to bile acids and increased bile acid synthesis in the liver. Colesevelam is a second-generation bile acid sequestrant that is used for lipid lowering in combination with statins, and with antidiabetic drugs for increasing glycemic control and insulin resistance [26,27]. Similar to our studies using fexaramine and FGF15/19, we did not observe a lowering of systemic bile acids in ethanol-fed mice following colesevelam treatment. However, colesevelam reduced hepatic steatosis by lowering liver triglycerides and cholesterol, similar to the effect we observed with fexaramine and FGF15/19, indicating that colesevelam might change the bile acid composition and signaling. Indeed, colesevelam as a resin stimulates bile acid signaling via the Takeda G protein-coupled receptor 5 (TGR5); consequently, colesevelam suppresses hepatic glycogenolysis by the TGR5-mediated induction of glucagon-like peptide 1 (GLP-1) [28]. In addition, we found that the expression of genes involved in *de novo* lipogenesis and cholesterol synthesis was reduced following colesevelam treatment in ethanol-fed mice, which likely contributes to decreased hepatic triglycerides and cholesterol levels.

Colesevelam improved serum bile acid levels in other mouse models such as *Mdr2*^−/−^ mice, used as a model for primary sclerosing cholangitis. Eight weeks of colesevelam treatment in *Mdr2*^−/−^ mice has been shown to attenuate liver and bile duct injury with the reduction of serum liver enzymes, bile acids, liver inflammation, and fibrosis [22]. In the same line, different clinical studies have denoted the potential of colesevelam to interrupt the enterohepatic circulation of bile acids and cause increased conversion of hepatic cholesterol to bile acids in different diseases, such as non-alcoholic fatty liver disease, cholestatic pruritus, Crohn’s disease, hypercholesteremia, and type 2 diabetes mellitus [17,18,20,29,30]. However, colesevelam increased serum ALT levels in ethanol-fed mice. One possible explanation is that colesevelam as a resin might not only bind bile acids, but possibly other beneficial molecules as well. We have previously demonstrated that the complete absence of microbes makes germ-free mice more susceptible to acute ethanol-induced liver injury [31]. Indeed, when mice fed a different bile acid sequestrant, cholestyramine, were challenged with acetaminophen, they showed impaired hepatic glutathione regeneration capacity and worsened liver injury [32].

In the present study, germ-free mice were colonized with stool samples from patients with alcoholic hepatitis and subjected to the chronic-binge ethanol feeding model [4,23]. Consistent with our previous studies [23], humanized mice developed liver injury, characterized by increased serum ALT levels and liver fat accumulation, and by increased hepatic levels of triglycerides and cholesterol. Non-humanized mice have increased hepatic Cyp7a1 expression following chronic ethanol feeding [14]. In contrast, hepatic Cyp7a1 was slightly, but not significantly, increased in gnotobiotic mice colonized with stool from alcoholic hepatitis patients, which is more similar to patients with alcohol-related steatohepatitis and alcoholic hepatitis [13]. Under normal chow-fed conditions, colesevelam binds bile acids in the intestinal lumen and increases fecal elimination. This will increase hepatic Cyp7a1 to counteract the fecal loss of bile acids. However, Cyp7a1 is not only regulated via the gut–liver axis, but also on the level of the liver. One primary inducer of Cyp7a1 is cholesterol via activation of liver x receptor (LXR)-alpha [33]. Since colesevelam reduced hepatic cholesterol in ethanol-fed mice, this might contribute to lower Cyp7a1 expression in ethanol-fed mice.

In conclusion, colesevelam reduces hepatic lipid accumulation, the hallmark of chronic ethanol feeding, but does not reduce liver injury in a gnotobiotic mouse model of chronic-plus-binge ethanol feeding.

## Figures and Tables

**Figure 1 cells-10-01496-f001:**
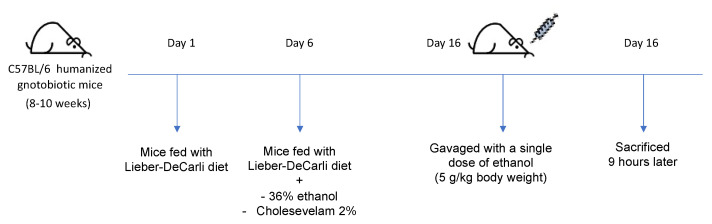
Experimental design.

**Figure 2 cells-10-01496-f002:**
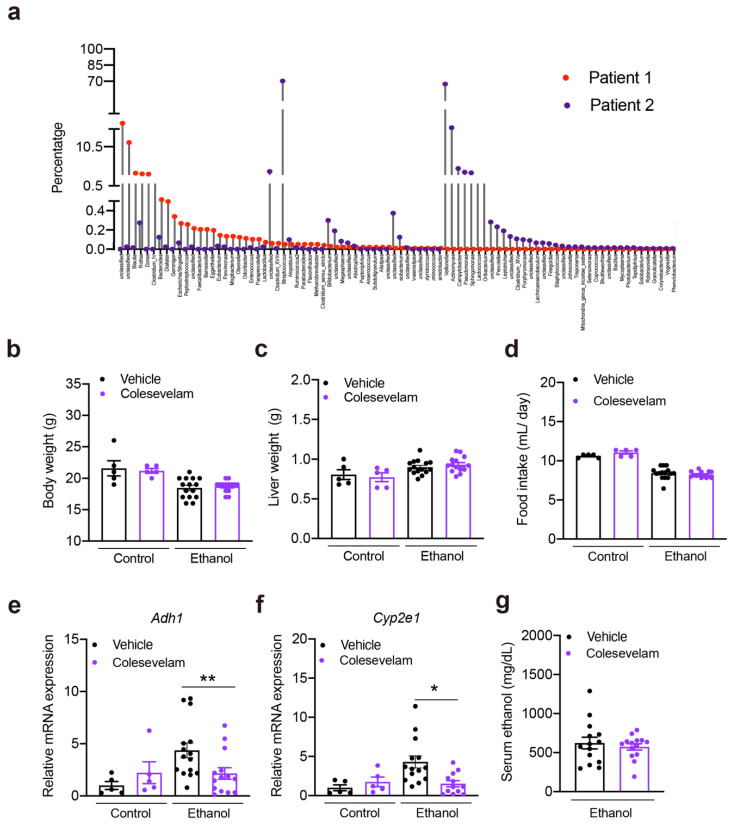
Colesevelam administration in gnotobiotic mice subjected to the chronic-plus-binge ethanol feeding model. C57BL/6 germ-free mice were colonized with feces from two patients with alcoholic hepatitis and subjected to the chronic-binge ethanol feeding model (*n* = 15 in vehicle and *n* = 14 colesevelam treatment) or isocaloric diet (*n* = 5 per group), and given vehicle or colesevelam in the liquid diet. (**a**) 16S rRNA analysis of stool from the two patients with alcoholic hepatitis. (**b**) Body weight. (**c**) Liver weight. (**d**) Food intake. (**e**,**f**) Hepatic *Adh1* and *Cyp2e1* mRNAs. (**g**) Serum level of ethanol. Results are expressed as mean ± s.e.m. *p* values were determined by one-way ANOVA with Tukey’s post-hoc test. * *p* < 0.05, ** *p* < 0.01.

**Figure 3 cells-10-01496-f003:**
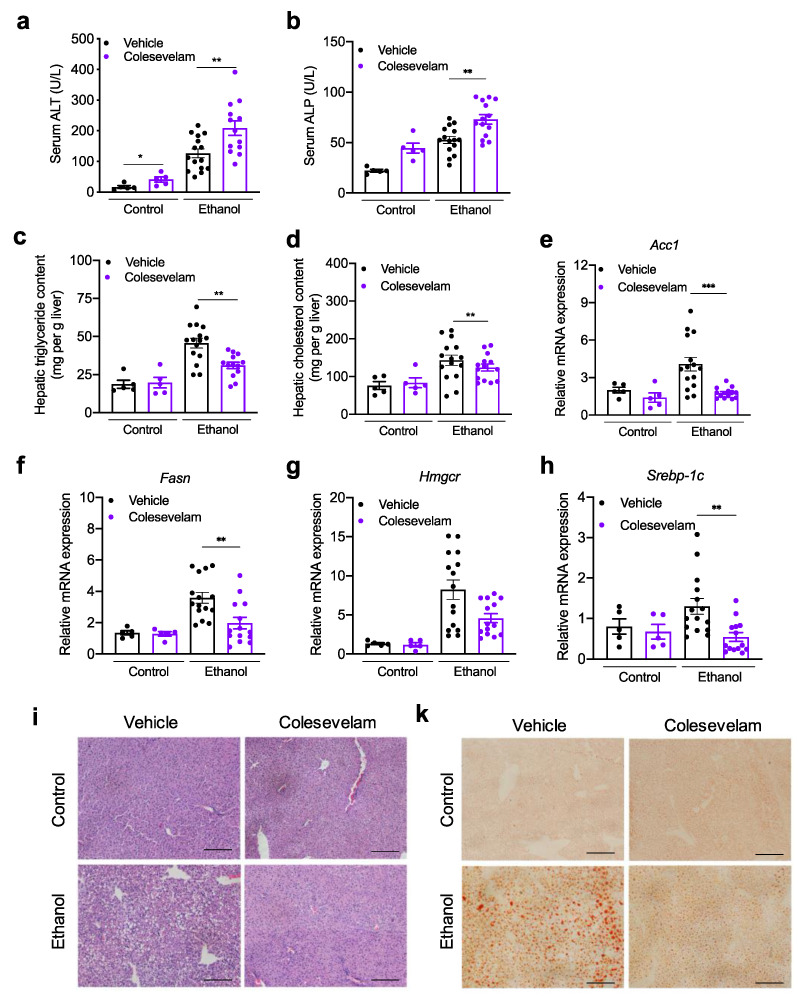
Colesevelam treatment ameliorates ethanol-induced liver steatosis in gnotobiotic mice subjected to the chronic-plus-binge ethanol feeding model. C57BL/6 germ-free mice were colonized with feces from two patients with alcoholic hepatitis and subjected to the chronic-binge ethanol feeding model (*n* = 15 in vehicle treatment and *n* = 14 colesevelam treatment) or isocaloric diet (*n* = 5 per group), and given vehicle or colesevelam in the liquid diet. (**a**) Serum level of alanine aminotransferase (ALT). (**b**) Serum level of alkaline phosphatase (ALP). (**c**) Hepatic triglyceride content. (**d**) Hepatic cholesterol content. Hepatic *Acc1* (**e**), *Fasn* (**f**), *Hmgcr* (**g**), and *Srebp1c* mRNA (**h**). (**i**) Representative liver sections after hematoxylin and eosin staining. (**k**) Representative Oil Red O-stained liver sections. Results are expressed as mean ± s.e.m. *p* values were determined by one-way ANOVA with Tukey’s post-hoc test. ** *p* < 0.01, *** *p* < 0.001. Bar size = 100 μm.

**Figure 4 cells-10-01496-f004:**
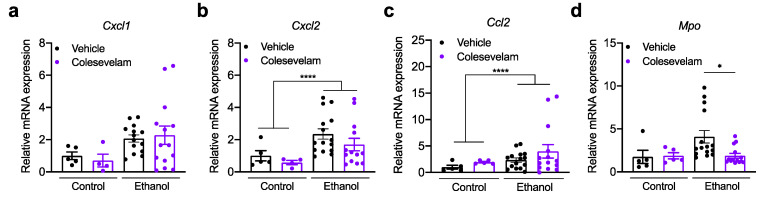
Effect of colesevelam on hepatic inflammation in gnotobiotic mice subjected to the chronic-plus-binge ethanol feeding model. C57BL/6 germ-free mice were colonized with feces from two patients with alcoholic hepatitis and subjected to the chronic-binge ethanol feeding model (*n* = 15 in vehicle and *n* = 14 colesevelam treatment) or isocaloric diet (*n* = 5 per group), and given vehicle or colesevelam in the liquid diet. Hepatic *Cxcl1* (**a**), *Cxcl2* (**b**), *Ccl2* (**c**), and *Mpo* mRNA (**d**). Results are expressed as mean ± s.e.m. *p* values were determined by two-way ANOVA with Tukey’s post-hoc test. * *p* < 0.05, **** *p* < 0.0001.

**Figure 5 cells-10-01496-f005:**
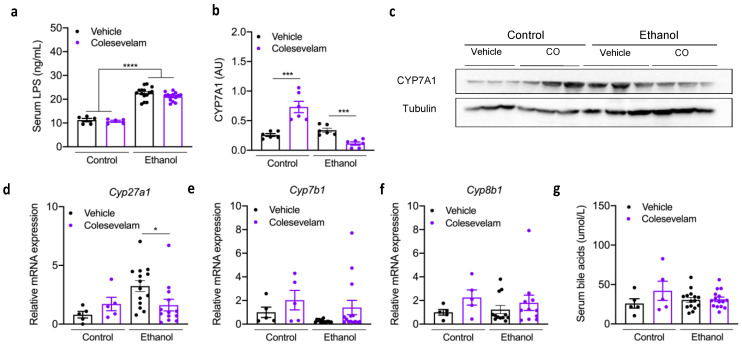
Effect of colesevelam treatment on intestinal barrier function and bile acid metabolism in gnotobiotic mice subjected to the chronic-plus-binge ethanol feeding model. C57BL/6 germ-free mice were colonized with feces from two patients with alcoholic hepatitis and subjected to the chronic-binge ethanol feeding model (*n* = 15 in vehicle and *n* = 14 colesevelam (CO) treatment) or isocaloric diet (*n* = 5 per group), and given vehicle or colesevelam in the liquid diet. (**a**) Serum LPS measured by ELISA. (**b**,**c**) Immunoblot analysis of hepatic Cyp7a1. Hepatic expression of *Cyp27a1* (**d**), *Cyp7b1* (**e**), *Cyb8b1* mRNA (**f**). (**g**) Plasma bile acid levels. Results are expressed as mean ± s.e.m. *p* values were determined by two-way ANOVA (**a**) or one-way ANOVA (**b**–**g**) with Tukey’s post-hoc test. * *p* < 0.05, *** *p* < 0.001, **** *p* < 0.0001.

**Table 1 cells-10-01496-t001:** Characteristics of patients with alcoholic hepatitis used as stool donors for germ-free mice.

Variables	Alcoholic Hepatitis	
	Patient 1	Patient 2
Sex (male)	Male	Male
BMI (kg/m^2^)	26.7	27.5
Age (years)	53	35
ALT (IU/L)	24.0	49.0
AST (IU/L)	84	170
ALP (IU/L)	231	116
Bilirubin (mg/dL)	14.1	8.0
Creatinine (μmol/L)	3.1	3.3
Albumin (g/dL)	2.4	1.8
INR	1.6	1.8
Platelet count (10^9^/L)	161.0	73.0
Stage of fibrosis	2	2
MELD	33	33
Child–Pugh stage	C	C

BMI: body mass index; AST: aspartate aminotransferase; ALP: alkaline phosphatase; ALT: alanine aminotransferase; INR: international normalized ratio; MELD: model for end-stage liver disease. Fibrosis is based on a clinically indicated liver biopsy using the Metavir scoring system.

**Table 2 cells-10-01496-t002:** Primers used in this study.

Gene	Primer	Sequence
Mouse 18S	F	5′-AGTCCCTGCCCTTTGTACACA-3′
	R	5′-CGATCCCAGGGCCTCACTA-3′
Mouse *Acc1*	F	5′-TGGAGAGCCCAACACACA-3′
	R	5′-GGACAGACTGATCGCAGAGAA-3′
Mouse *Adh1*	F	5′-GGGTTCTCAACTGGCTATGG-3′
	R	5′-ACAGACAGACCGACACCTCC-3′
Mouse *Cxcl1*	F	5′-TGCACCCAAACCGAAGTC-3′
	R	5′-GTCAGAAGCCAGCGTTCACC-3′
Mouse *Cxcl2*	F	5′-AAAGTTTGCCTTGACCCTGAA-3′
	R	5′-CTCAGACAGCGAGGCACATC-3′
Mouse *Ccl2*	F	5′-ATTGGGATCATCTTGCTGGT-3′
	R	5′-CCTGCTGTTCACAGTTGCC-3′
Mouse *Cyp2e1*	F	5′-GGGACATTCCTGTGTTCCAG-3′
	R	5′-CTTAGGGAAAACCTCCGCAC-3′
Mouse *Cyp7b1*	F	5′-AGGCATGACGATCCTGAAATAG-3′
	R	5′-CAGCCTCAGAACCTCAAGAATAG-3′
Mouse *Cyp8b1*	F	5′-AGGGTGGTACAGGAGGATTAT-3′
	R	5′-GAGTCTGAGCTGGTAGGATTTG-3′
Mouse *Cyp27a1*	F	5′-GGAGGGCAAGTACCCAATAAG-3′
	R	5′-GCAAGGTGGTAGAGAAGATGAG-3′
Mouse *Fasn*	F	5′-GTCGTCTGCCTCCAGAGC-3′
	R	5′-GTTGGCCCAGAACTCCTGTA-3′
Mouse *Hmgcr*	F	5′-GGCCTCCATTGAGATCCG-3′
	R	5′-CACAATAACTTCCCAGGGGT-3′
Mouse *Mpo*	F	5′-GATGACCCCTGCCTCCTC-3′
	R	5′-GCTCTCGAACAAAGAGGGTG-3′
Mouse *Srebp1c*	F	5′-GGAGCCATGGATTGCACATT-3′
	R	5′-GCTTCCAGAGAGGAGGCCAG-3′

## Data Availability

Raw 16S sequence reads have been published [23] and can be found in the NCBI SRA associated with Bioproject PRJNA525701.

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
