# Peer review of "Colesevelam Reduces Ethanol-Induced Liver Steatosis in Humanized Gnotobiotic Mice"

_cells, 2021, doi:10.3390/cells10061496_

Round 1

Reviewer 1 Report

The review of the manuscript title: “Colesevelam reduces ethanol-induced liver steatosis in humanized gnotobiotic mice” authored by Cabré et al.

The revised manuscript aimed to assess the potential role of the bile acid sequestrant colesevelam in a humanized mouse model of ethanol-induced liver disease. The manuscript is interesting and provides valuable contribution of the basic research in field of the therapy of alcohol liver disease.

Despite the fact, that he manuscript is scientifically very valuable, it requires some improvement before consideration of publication in Cells journal.

My decision: major revision

Please follow my specific comments:

  • All the abbreviations should be explained when are mentioned for the first time!
  • Introduction: An issue regarding a humanized mouse model of ethanol-induced liver disease should be extended;
  • Introduction: The aim of the study should be clearly indicated and should refer to the novelty of the study;
  • Table 2 should move to the supplementary material;
  • Materials and methods: an additional information regarding the assay kits should be provided to the manuscripts (e.g., intra- and inter-assay precision, linearity, the level of sensitivity etc.)
  • Materials and methods: the graphical scheme of experimental design is highly recommended;
  • Materials and methods: please include a brief protocol of immunoblot analysis;
  • In my opinion the control animals that were not colonized with human feces and exposed to the ethanol Lieber-DeCarli diet are needed to confirm ALD in mice.

Author Response

Please see the attachment we provide a point-by-point all the comments.  

Reviewer 2 Report

Cabre N et al. reported the potential role of the bile acid sequestrant colesevelam in a humanized mouse model of ethanol-induced liver disease. Ethanol-fed gnotobiotic mice treated with colesevelam showed reduced hepatic levels of triglycerides and cholesterol, but liver injury and inflammation were not decreased as compared with non-treated mice. Colesevelam reduced hepatic Cyp7a1 protein expression, although serum bile acids were not lowered. Colesevelam failed to reduce alcoholic liver disease. Although it reduced hepatic triglycerides and cholesterol, the mechanism of this reduction has not been analyzed at all. This reviewer thinks that not so much valuable information will be included in this manuscript. Followings are my comments to the authors.

  1. Was the duration of treatment with Colesevelam appropriate? The authors used the NIAAA model to create alcoholic liver injury.1 This model can be extended to longer periods of chronic feeding (up to 8 weeks) plus single or multiple binges.1 However, the authors did not state the duration of ethanol administration or the total duration of Colesevelam administration. Fuchs CD et al. treated Mdr2-/- mice with colesevelam for 8 weeks.2 This reviewer thinks that Colesevelam should be given for at least 8 weeks.

  1. The mechanism of triglyceride and cholesterol reduction in the liver has not been analyzed. The authors should conduct an experimental study to determine why Colesevelam reduces intrahepatic triglycerides and cholesterol in patients with alcoholic liver disease.

  1. Table 2.
  2. The % should be deleted.
  3. The BMI units are missing.
  4. The authors should describe the age of the patient and daily alcohol intake.

  1. Figure 2. The authors should include data on serum alkaline phosphatase concentrations. The authors should also examine gene expression in the synthetic and scavenging systems of triglycerides and cholesterol in the liver.

  1. Figure 2-d and e. The quality of the photos is extremely poor. Especially Oil red-O staining is impossible to judge in this photo. Scale bar is also lacking.

  1. Figure 3. For pro-inflammatory cytokines and chemokines, mRNA expression of TNF-alpha, IL-6, and MCP-1 should also be examined. Also, please describe the data of F4-80 immunostaining.

  1. Figure 4-b. Why was CYP7A1 increased by Colesevelam in the control group and decreased in the ethanol group? The authors should discuss this in detail, including the mechanism.

References:

  1. Bertola A, Mathews S, Ki SH, Wang H, Gao B. Mouse model of chronic and binge ethanol feeding (the NIAAA model). Nat Protoc. 2013;8(3):627-637.
  2. Fuchs CD, Paumgartner G, Mlitz V, et al. Colesevelam attenuates cholestatic liver and bile duct injury in Mdr2(-/-) mice by modulating composition, signalling and excretion of faecal bile acids. Gut. 2018;67(9):1683-1691.

Author Response

(The authors gave the same response as above.)

Reviewer 3 Report

Topic: higly prevalent disease, therapy unemt need = of high interest

Abstract: clear, dense

English: OK

Introduction: setting the stage is appropriate

Aim: missing (replaced by "we assessed"). It would be helpful if authors took the pains of clearly formulating their hypothesis behind the endeavour. If it is left to a less informed reader, the results may well be understood as negative.

Methods: state-of-the art as expected from such an eminent senior author. The only info I am missign is on the fecal samples from the patients: a)were they mixed together prior to administration? (i.e. doses were "equipotent") and b)were the microbiomes of the two phenotypically quite different patients  examined and if yes were there any differences relevant to the scope of the study?  

Results: Clear, sufficient for conslusions.

Table 2. Please add to footnote of the Tab the staging system according to which the fibrosis was classified and what was the diagnostic method.

Figure 2. Please help reader in searching for "find differences" in sections d) and e) by brief description, e.g. "notable is the increase in ..." 

Discussion:

Please provide your more elaborate explanations for these very interesting findings:

  • colesevelam reduced expression of ethanol-metabolizing enzymes but there was no difference in serum ethanol
  • colesevelam increased ALT:  (equal ethanol, less steatosis, higher ALT).  Since it is one of the core findings of the study, it deserves more than single-sentence statement and one reference (30). Please add your / most probable explanation behind "resins are associated": what is the mechanism? what are the consequences? is there any hint in your histological findings?
  • colesevelam increased CYP7A1 in controls but decreased its expression in ethanol-fed mice

Author Response

(The authors gave the same response as above.)

Round 2

Reviewer 1 Report

The Authors have properly addressed all my comments and suggestion. I recommend the manuscript for publication in the "Cells" journal.

Reviewer 2 Report

The peer reviewers' advice made this paper very good. I think it should be accepted.